# Studying Individual Differences in Spatial Cognition Through Differential Item Functioning Analysis

**DOI:** 10.3390/brainsci10110774

**Published:** 2020-10-24

**Authors:** Antonella Lopez, Alessandro O. Caffò, Luigi Tinella, Albert Postma, Andrea Bosco

**Affiliations:** 1Department of Educational Sciences, Psychology, Communication, University of Bari, 70121 Bari, Italy; alessandro.caffo@uniba.it (A.O.C.); luigi.tinella@uniba.it (L.T.); andrea.bosco@uniba.it (A.B.); 2Helmholtz Institute, Experimental Psychology, Utrecht University, 3584 CS Utrecht, The Netherlands; A.Postma@uu.nl

**Keywords:** individual differences, sketch map, familiar environments, psychometrics, binary differential item functioning (DIF), categorical and coordinate spatial relations

## Abstract

*Background*: In the field of spatial cognition, the study of individual differences represents a typical research topic. Gender and age have been prominently investigated. A promising statistical technique used to identify the different responses to items in relation to different group memberships is the Differential Item Functioning Analysis (DIF). The aim of the present study was to investigate the DIF of the Landmark positioning on a Map (LPM) task, across age groups (young and elderly) and gender, in a sample of 400 healthy human participants. *Methods*: LPM is a hometown map completion test based on well-known and familiar landmarks used to assess allocentric mental representations. DIF was assessed on LPM items two times: on categorical (i.e., positions) and coordinate (i.e., distances) scores, separately. *Results*: When positions and distances were difficult to assess with respect to the intended reference point, the probability to endorse the items seemed to get worse for the elderly compared to the younger participants. Instead other features of landmarks (high pleasantness, restorativeness) seemed to improve the elderly performance. A gender-related improvement of probability to endorse distance estimation of some landmarks, favoring women, emerged, probably associated with their repeated experiences with those landmarks. Overall, the complexity of the task seemed to have a differential impact on young and elderly people while gender-oriented activities and places seemed to have a differential impact on men and women. *Conclusions*: For the first time DIF was applied to a spatial mental representation task, based on the schematic sketch maps of the participants. The application of DIF to the study of individual differences in spatial cognition should become a systematic routine to early detect differential items, improving knowledge, as well as experimental control, on individual differences.

## 1. Introduction

The study of individual differences represents a major topic in psychological sciences. Individuals show differences in behavior, cognition, personality, and attitudes, arising from biological as well as cultural and socio-economic factors. In spatial cognition as well, the study of individual differences has represented a classical research line. Gender and age have received greater attention with respect to other variables. Regarding gender differences, several reviews and meta-analyses witnessed the amount of primary studies on the role of gender in spatial abilities (e.g., [1,2,3,4,5]), mental rotation [6,7,8], spatial orientation [9], spatial navigation [10], spatial learning and memory in cognition [11] and sports [12], categorical and coordinate spatial relations [13], visuo-spatial abilities in adults [14] and in students with learning disabilities [15], and spatial reasoning [16]. Age differences have also been investigated and summarized in reviews and meta-analyses concerning spatial cognition in general [17,18] spatial memory [19] and spatial navigation processes in both normal [20] and impaired aging [21], egocentric and allocentric spatial reference frames in aging [22] virtual reality for the diagnosis of spatial navigation disorders [23], visuospatial working memory [24], topographical disorientation in aging within familiar and unfamiliar environment [25], and heritability of cognitive aging [26]. Dealing with gender and age differences in assessment and intervention contexts poses at least two complementary issues, one in which researchers and professionals are interested in finding and employing tools that can dissolve such differences, and the other in which they are interested in finding and employing tools that can maximize them. Sometimes the aim is to rely on a tool to assess a construct without worrying about a potential differential effect due to individual differences, and some other times the aim is to select a tool suitable to assess a construct in a specific group only (i.e., women rather than men, or elderly rather than young people). Moreover, when creating a psychological test, it is a rare eventuality to think of different forms suitable to specific groups before a general validation has occurred. In a similar vein, it is equally rare to obtain invariance across given groups without modifying the structure of a test, by adding or deleting items.

A statistical and psychometrics technique used to identify tests whose scores may be affected by groups’ characteristics is Differential Item Functioning (DIF) analysis. DIF occurs when groups show different probabilities of endorsing a single item on a multi-item scale after controlling for overall scale scores. An item is considered as having DIF when individuals with the same ability but coming from different groups show an unequal probability of giving a response. An item is considered as non-DIF when individuals with the same competence have equal probability of giving a correct response, irrespective of group membership. It is worthwhile to note that DIF can be used for measurements across both groups times. Several data analytic strategies have been adopted for detecting DIF, such as the Mantel-Haenszel approach, the odds ratios, Item Response Theory models, and logistic regression. The latter method is widespread for the following reasons: first, several standardized tests or experimental trials and measures use questions which are or can be dichotomized as “correct” or “incorrect”, and logistic regression represents the natural strategy to analyze binary outcomes (e.g., [27,28]). Second, logistic regression can detect both uniform and non-uniform DIF employing two or more categorical groups or a continuous random variable, thus allowing for a wide applicability for most psychological measures. Third, it can be used on single items of a test to retain or remove one or more items. Indeed, logistic regression for detecting DIF involve running a separate analysis for each item. The independent variables are an ability matching variable, usually a total score of a test, group membership, and an interaction term between the two former variables. The dependent variable is the probability or likelihood of endorsing an item or getting a correct response. The logistic regression equation used to detect DIF is the following:Y = β0 + β1M + β2G + β3MG
where β0 corresponds to the intercept or the probability of a response when M and G are equal to 0, β1 corresponds to weight coefficient for the first independent variable, M, namely the matching variable used to link individuals on ability, β2 corresponds to weight coefficient for the second independent variable, G, namely the group membership variable, and β3 corresponds to weight coefficient for the interaction term between the two above mentioned variables, MG. Variables are entered hierarchically by the following sequence: matching variable M, grouping variable G, and the interaction variable MG. 

DIF analysis has been proficiently used in several psychological fields, such as social (e.g., [29,30,31]), clinical (e.g., [32,33]), and cognitive (e.g., [34,35]) psychology. Its use seems not to be confined to paper-and-pencil self-reported questionnaires only, but there also is a large employment for behavioral (e.g., [36]), neuropsychological and performance-based tools (e.g., [37,38,39]). To the best of our knowledge, there are no studies employing DIF analysis in spatial cognition performance measures, except for Choo, Park, and Nelson [40] which evaluated the impact of mild learning disabilities on students’ spatial thinking skills.

In the present study, DIF analysis based on logistic regression will be applied to a task developed by Lopez et al. [25] to investigate hometown topographical knowledge. Recent literature showed that familiarity with the environment might represent a protective factor from spatial memory impairment in non-pathological cognitive aging, in particular for allocentric tasks. Indeed, information consolidated across a huge number of retrieval episodes seems to be solidly preserved in elderly people, making them to obtain a performance comparable with that of younger people [41,42,43]. The Landmark Positioning on a Map (LPM, [25]) is used to assess allocentric remote spatial memory, in which participants had to identify and pinpoint the correct position of eight well-known landmarks of their hometown on a blind map of the city, keeping in mind metric (i.e., relative distances) as well as categorical (“A is above/below and left/right of B”) spatial relationship between landmarks. LPM is intended to be age and education fair, as it did not favor young people compared with aged ones or healthy elderly with high and low level of education [44]. In the study, the group dependent contribution of a single item to the total score of the test is not known. There could be some items which elicit different performances favoring a specific group of participants, for example men over women or young over elderly people, other items which favor the other group, and still others which do not show any difference in functioning between groups.

In this view, the theoretical and practical relevance for using DIF analysis in the field of spatial cognition and in particular in spatial memory based on familiar topographical information is given by, for instance, the selection of items which can be representative and fair for many groups of people. Such selection could stimulate the construction of standardized tools, even within an ecological approach, to study spatial mental representations, where it could be difficult to control for previous learning processes, strength and accessibility of memory trace, and, as such for individual differences. DIF analysis allows to identify the differential functioning between groups. In this framework, it is then possible (a) to remove the unfair landmarks with respect to different groups or (b) to select landmarks which are appropriate for a specific group only. It is worthwhile to note that DIF analysis do not tell anything *per se* about the fairness or the biasedness of a test and, consequently, the eventual choice to exclude/substitute an item is demanded to the researcher.

The aim of the present study is twofold: (1) to investigate DIF on the landmarks of a test of familiar spatial mental representations in relation to age and gender groups, and (2) to investigate whether DIF is present specifically on categorical (i.e., positions) and coordinate (i.e., distances) scores.

## 2. Materials and Methods

### 2.1. Participants

A power analysis to estimate the required sample size was carried out using G*Power 3.1 [45]. The sample size was established by considering three aspects: a level of significance equal to 0.05; a power of 0.95; and an odd ratio of 1.5. As a result, a sample size of 417 participants was sufficient to warrant that the probability of correctly rejecting the H0 hypothesis was 95%, indicating that there is no relation between the main predictor and result variable. 

Four hundred and twenty participants were recruited. Twenty elderly participants were excluded because their global cognitive functioning, assessed by the Montreal Cognitive Assessment [46,47], fell under the normality criterium (>17 points; [48]). At the end of the enrolment procedure, four hundred healthy participants (201 women) took part in the study. All participants were from the metropolitan area of Bari (Apulia, Italy). Two hundred and three young university students (i.e., age mean ± sd 23.61 ± 5.53; level of education mean ± sd 16.17 ± 1.61), and 197 elderly people (i.e., age mean ± sd 73.06 ± 6.80; level of education mean ± sd 11.47 ± 5.08) were enrolled in the study. Descriptive statistics for the two groups are reported in Table 1. All participants, blind to the hypothesis of the study, signed a consensus form. The local ethical committee (No. 3660-CEL03/17 November 2017) approved the study protocol, and the whole study was performed following the Helsinki Declaration and its later amendments. 

### 2.2. Materials and Procedure

#### Hometown Task: Landmark Positioning on a Map

Participants were required to complete landmark positioning on a map task (LPM, [25], see Figure 1. Participants were first required to recognize 10 well-known landmarks in their hometown that were displayed in photographs (see Figure 2a,b). The given references for the hometown map were the marking for North, and two of the 10 landmarks, namely one in the centre of a semi-blind map and the other further outside the city on the map. The two landmarks served as positional and distance reference points which allowed participants to infer positions and distances of the other landmarks. The participant had to pinpoint all the other eight landmarks, keeping in mind the metric (i.e., relative distances) as well as categorical (relative positions) spatial relations between landmarks (see Figure 2c). The LPM task is known to be relatively unbiased with respect to age and education, and relatively sensitive in discriminating between different levels of cognitive functioning [44].

Participants were assessed individually in a well-lit, quiet room without disturbances. Each step in the testing procedure was made clear to the participants beforehand. Data were collected in one session ranging between 15–30 min. Breaks were allowed on request.

### 2.3. Scoring Method

Categorical and coordinate scores were computed. Categorical and coordinate spatial relations could be considered as latent variables. They can be operationalized according to two empirical measures, namely relative positions, and distances, respectively.

In order to evaluate the landmarks positions’ the following procedure was followed: the target area was divided into four quadrants, starting from the origin (Bari Central Railway Station, Landmark 1, in Figure 2), namely the point where a two-axes, east-west and north-south, intersect. For each correct position, participants were awarded of 1 point, conversely 0 was assigned for incorrect positions, regardless of the distance from the origin. 

In order to evaluate distances, a circle, whose origin coincided with Bari Central Railway Station, was drawn with a radius equal to the median distance from the centre of the circle to each landmark. In this way, four *far* and four *near* landmarks were identified. Each positioning was awarded of 1 when the reference point was positioned at the correct distance from the origin, and of 0 vice versa, regardless of the position. 

### 2.4. Statistical Analysis

The analyses focused on evaluating binary differential item functioning (DIF) of LPM Task across age group (young and elderly participants), and gender. DIF analysis seeks to find out if the young and the elderly, and men and women, were equally proficient, that is had an equal probability of positioning a given item (landmark) correctly. In this case two logistic regressions were used to detect both uniform and non-uniform DIF with two categorical variables: age and gender. Uniform DIF is a pattern of DIF where the degree of DIF is constant across the spectrum of ability being assessed by a test, whereas non-uniform DIF acts as a function of both the degree of ability and the grouping variable. Logistic regression is considered the most flexible method of detecting DIF (for more details, [49,50]). Age group and gender were considered as the grouping variables (G) with a binary classification 0/1 (young and elderly participants and women and men, respectively). The general categorical and coordinate scores were considered as the matching variables (θ, or latent ability variable). Using jamovi [51] as a statistic software, which facilitates the reporting of DIF analyses, DIF computation involved the estimation of three models and the comparison of these models’ fit. The first model is based on calculation of the main effect of the matching variable; the second one contains the main effect of the matching variable and the main effect of the grouping variable, and the third model contains also an interaction term between the matching and the grouping variable. A chi-square test with two degrees of freedom between these three models is performed to detect DIF, which is present if individuals matched on ability show significantly different probabilities of responding to an item, and thus differing logistic regression curves. On the contrary, DIF is not present if curves for both groups are the same, and then the item is considered as unbiased. Finally, there is evidence of uniform DIF if the intercepts and matching variable parameters for both groups are not equal, while there is evidence of nonuniform DIF if there is a nonzero interaction parameter [52]. If the change in model fit is not statistically significant, the item is not flagged as exhibiting DIF. Otherwise, if the change in model fit is significant, the item is flagged as exhibiting DIF. Uniform and non-uniform types of DIF were assessed for checking simultaneously for significant main and interaction effects. Wald and LTR (Likelihood Ratio Test)as Flagging Criteria (tests used for assessing statistical significance of regression coefficients) and Zumbo-Thomas (ZT) and Jodoin-Gierl (JG) as Evaluation Scale (classification scheme for DIF effect sizes on the Naeglekirke’s *R*^2^ (Δ*R*^2^) scale) were calculated. Items flagged, as exhibiting DIF, were evaluated using the following effect sizes scale for ZT method: “‘A”: Negligible effect (0 ≤ Δ*R*^2^ ≤ 0.13),“B”: Moderate effect (0.13 ≤ Δ*R*^2^ ≤ 0.26), and “C”: Large effect (0.26 ≤ Δ*R*^2^ ≤ 1); for JG method: “A”: Negligible effect (0 ≤ Δ*R*^2^ ≤ 0.035), “B”: Moderate effect (0.035 ≤ Δ*R*^2^ ≤ 0.07), and “C”: Large effect (0.07 ≤ Δ*R*^2^ ≤ 1). JG method has a less conservative assessment criterion, compared to ZT method. To the study objectives, ZT effect sizes were reported and interpreted.

Items flagged should have *p*-values’ of the χ^2^ statistic associated with the standard 2-degrees of freedom for Likelihood Ratio/Wald test below the chosen threshold of 0.05. In the present research the item flagged were discussed, regardless small, moderate, or large effect size. The binary logistic regression coefficients and the associated odds were used to interpreted DIF, in terms of the increased likelihood of response between groups. The item response curves were plotted for any item being assessed for DIF. The curves provide a visual representation of the full model used in the DIF analysis. The different groups are colour coded, with 95% confidence intervals represented by gray shading overlaying each line. The *Y*-axis of the plot is the probability (from 0 to 1) of correct positioning (categorical component) and correct estimation of distance with respect to Bari Central Railway Station (coordinate component), and the *X*-axis is the range of the matching variable (i.e., the latent ability measure) used in the logistic models. Line graphs illustrate the trends of the probability of correct item positioning or distance estimation from Bari Central Railway Station, by the participants. If the lines are parallel to *x*-axis, no change in slope over the spectrum of θ (test taker proficiency) is exhibit, instead, if the lines are not parallel, crossing one another, the plot indicates that difficulty of the item changes as a function of both group membership and θ level (e.g., [53]).

## 3. Results

Descriptive statistics and preliminary analyses on categorical and coordinate scores are reported in Table 1. 

### 3.1. Grouping Variable: Age 

Considering categorical scores as matching variable, four items of the LPM task were flagged (see Table 2). according to the Zumbo and Thomas (ZT) scale. The flagged items showed a A-Level DIF, as shown in Table 3.

Each item is discussed in turn, reporting also the plots showing the item characteristic curves (see Figure 3).

Item#1: Basilica di San Nicola

The age DIF test is statistically significant χ^2^ (2) = 11.06, *p* = 0.004. The Δ*R*^2^ = 0.037 is a small effect by ZT criterion. Both the significance test and the effect size suggested age DIF. As emerged from Table 3 the young were approximately 1.4 times more likely to correctly place item#1 than the elderly. Figure 3 shows that the probability of the young to locate the item increased at higher item response scores more than that of the elderly, whose probability remains the same for low, medium and high scores. In other words, the item was not discriminated by the elderly regardless the level of ability. 

Item#2: Casa Circondariale di Bari

The age DIF test is statistically significant χ^2^ (2) = 19.08, *p* < 0.001. The Δ*R*^2^ = 0.053 is a small effect by ZT criterion. Both the significance test and the effect size suggested age DIF. As emerged from Table 3 the young were uniformly 1.2 times more likely to correctly place item#2 than the elderly.

Item#4: Spiaggia “Pane & Pomodoro”

The age DIF test was statistically significant χ^2^ (2) = 11.48, *p* = 0.003. The Δ*R*^2^ = 0.041 is a small effect by ZT criterion. Both the significance test and the effect size suggest age DIF. As emerged from Table 3 the elderly were approximately 1.8 times more likely to correctly place item#4 than the young. Figure 3 shows that the probability of correctly positioning the item was greater for the elderly at low levels of ability, while it was greater for young people at high levels of ability.

Item#5: Parco II Giugno

The age DIF test is statistically significant χ^2^ (2) = 9.30, *p* = 0.010. The Δ*R*^2^ = 0.016 is a small effect by ZT criterion. Both the significance test and the effect size suggested age DIF. As emerged from Table 3 the elderly were approximately 1.4 times more likely to correctly place item#5 than the young. Figure 3 shows that at low levels of ability, the probability of correctly positioning the item was slightly lower for the elderly, while at high levels of ability the probability became lower for the young. Taking into account coordinate scores as matching variable, one item of LPM task (Item#4: Spiaggia “Pane & Pomodoro”) was flagged, and it exhibited A-Level DIF according to ZT scale. The results are summarized in Table 2. Item#4 is discussed below. The item characteristic curve was reported in Figure 3.

Item#4: Spiaggia “Pane & Pomodoro”

As shown in Table 4, the age DIF test is statistically significant χ^2^ (2) = 13.53, *p* = 0.001, Δ*R*^2^ = 0.067. Both the significance test and the effect size suggest age DIF. The young were approximately 1.5 times more likely to make a correct estimation of distance than the elderly. Figure 3 shows that young people had a higher probability to give a correct answer at high levels of ability. Conversely, at low levels of ability the probability became higher for the elderly. Item#4 seemed to cause confusion: the gray shading overlaying each line was very wide (high variability in the first part of the curve), showing that the item had not a reliable score.

### 3.2. Grouping Variable: Gender

Considering categorical scores as matching variable, only the item Parco II Giugno was flagged and exhibited A-Level DIF according to ZT scale. The results are summarized in Table 5. Each item is discussed in turn, reporting also the plots showing the item characteristic curves (see Figure 4).

Item#5: Parco II Giugno

As emerged from Table 6, the gender DIF test is statistically significant χ^2^ (2) = 6.86, *p* = 0.032, Δ*R*^2^ = 0.013. Both the significance test and the effect size indicate gender DIF. Men were approximately 1.8 times more likely to place item#5 than women. Figure 4 shows that at low levels of ability, the probability of correctly positioning the item was slightly lower for men, while for high levels of ability the probability became lower for women.

Considering coordinate scores as matching variable, two items were flagged. As shown in Table 7, Basilica di San Nicola (Item#1) and Stadio della Vittoria (Item#8) exhibited A-Level DIF according to ZT scale.

Item#1: Basilica di San Nicola

The gender DIF test is statistically significant χ^2^ (2) = 6.04, *p* = 0.049. The Δ*R*^2^ = 0.015 is a small effect by ZT criterion. Both the significance test and the effect size suggested gender DIF. As emerged from Table 7 women were approximately 1.7 times more likely to make a correct estimation of distance than men. Figure 4 shows that at low levels of ability, the probability of give the correct response was slightly higher for men, while at high levels of ability women had a significantly higher probability than men to correctly estimate distance. It is evident that the interaction is confined in the initial part of the distribution, showing preponderant uniform effect of the grouping variable, for the benefit of women.

Item#8: Stadio della Vittoria

The gender DIF test is statistically significant χ^2^ (2) = 9.87, *p* = 0.007. The Δ*R*^2^ = 0.070 is a small effect by ZT criterion. Both the significance test and the effect size suggest gender DIF. Ceiling effects emerged from Figure 4: the probability to correctly estimate distance was substantially equivalent between men and women, with an overall benefit for women (Uniform type of DIF).

## 4. Discussion 

Item analysis examines participant responses to individual test items (questions, trials, stimuli) to assess the effectiveness and reliability of those items and, in turns, of the test as a whole. Item analysis is especially valuable in improving items which will be used again in later assessment procedures, and it can also be employed to support the researcher decision in eliminating or substituting ambiguous or misleading items. The use of binary differential item functioning (DIF) gave the opportunity to arrange a study on individual differences, consisting of a detailed item characteristic analysis. Usually, in spatial cognition, researchers investigate internal consistency, reliability and construct validity of the tests, or individual differences on the complete scales, eluding the study of item characteristics. Instead, DIF analysis paves the way to two opposite aims: a) to identify differential item functioning to make the scale unbiased with respect to the intended variable, possibly replacing items, freeing the test from probable biases; and b) to identify differential item functioning to better discriminate between certain characteristics, such as, age, gender or, even better, between different cognitive styles (e.g., [54]).

For the first time, in the present study, binary differential item functioning (DIF) was applied to a spatial mental representation task, based on the schematic sketch map of the participants. LPM task is a hometown map completion test used to assess allocentric topographical disorientation, and to discriminate, in particular, typical from atypical aging [25,41,44]. The aim of the present study was to examine individual differences (age and gender-related) and to speculate why some items showed a spread between age or gender groups. We explored age and gender DIF, across young and elderly participants, and men and women, measuring the accuracy in placing each landmark of LPM task, in terms of both categorical and coordinate spatial relations. The interest of research in deepening the differences in the encoding of categorical and coordinate spatial relations has important practical implications for the assessment of topographical orientation along life span (e.g., [42,55]). From preliminary analysis, on the overall categorical score, age and gender differences did not emerge, while on the overall coordinate score age differences occurred, in favor of the young. Notwithstanding these global results, the expected direction of effects, on each item, were not present. It is evident that DIF analysis is useful for tasks or tests that show clear grouping effects, but even when grouping effect are not present, there is no guarantee that the items behave in a completely fair way, hiding unexpected effects, actually. Age and gender item-related differences were discussed below into details. 

### 4.1. Age Related Differences

The first research question concerns age-related differences in categorical and coordinate spatial relations. With respect to the categorical component, four items showed age differences. Item#1 and Item#2 showed a clear disadvantage for elderly participants. This evidence can be explained by the difficulty of the landmark placement process. The items are almost perfectly aligned to the fixed reference point, on the North–South axis. This result is compatible with previous findings of Costa and Bonetti [56] and Lopez and colleagues [44] using the Map of Italy and also with Dror and Kosslyn [57], Saimpont [58] and De Beni [59], with regard to the differential access to mental spatial representations of landmark placed in front or behind the intended place of observation (i.e., aligned and counter-aligned landmarks). On the other hand, the elderly showed a slight advantage, with respect to the young, in placing Item#4 and Item#5. It is likely that this advantage is attributable to their non-spatial features. They are visually prominent, structurally significant, and salient [60,61]. They are characterized by a certain shape (they are extended in the space), visible from the pathway, with a leisure meaning (a beach with a seaside pathway and the city largest urban park, respectively), fundamental qualities contributing to their strong place identity [62]. They can be qualified as visually, semantically, and structurally attractive [63] and they are frequented by the elderly as meeting and socialization places. These characteristics improve their memorability [64], and attenuate age-related memory deficits [65].

Only the Item#4 showed age differences in distance estimation from the reference point, in favour of the young. This landmark is particularly close to the circumference of the circle used as reference to score the participants’ performance.. Overall, elderly people seemed to suffer from the complexity of the task. Landmarks characterized by a hard discrimination, due to their alignment with the reference point or to the proximity to the circumference adopted as reference for distance calculation, contribute to the disadvantage for the elderly in particular in distance estimation (e.g., [66,67]).

### 4.2. Gender Related Differences

Regarding categorical scores, men seemed to have a higher probability to correctly place the Item#5 compared to the women. The park is used for street sport purposes from male population (basket and soccer for the youngest, and bowl games for the elderly). On the other hand, women seemed to be more accurate in the estimation of distances regarding Item#1 and Item#8. Concerning Item#1, it is known that there is a gender gap in attendance at places of worship [68]. Moreover, Italian women report higher rates of weekly church attendance than men [69]. Item#8 is the old stadium of the city of Bari withdrawn in 1990. Currently, it hosts children’s theatre, and it is near to a pool center frequented predominantly by women (i.e., for the availability of aqua fitness and prenatal and childbirth classes). Moreover, the parental role with children, in holding daily activities including sports or culture, is carried out mainly by women (e.g., [70]). Hence, it is more likely that women frequented those places more than men.

This study has a potential limitations: the DIF effects, according to Zumbo-Thomas scale, can be considered small, nonetheless it can be considered significant and potentially informative regarding the contribution of each item to the differences between levels of grouping variables. 

## 5. Conclusions 

The investigation of differential item functioning (DIF) is a key methodology to show measurement similarities across different groups with respect to a certain feature [32]. Concerning the study of individual differences in spatial cognition, differences may arise in behavior, cognitive style, cognition, personality, attitudes, and demographic characteristics e.g., [11,31,71,72]. So, the use of DIF could make a substantial contribution to ensure a valid interpretation of group differences in those variables that affect spatial cognition.

Moreover, the relapses from the clinical and applicative point of view (i.e., driving assessment, personality evaluation) should be taken into consideration. A statistically significant DIF could give the clinician the opportunity to establish if an item could be considered clinically meaningful, and to be used for diagnostic purposes, and in all those assessment and evaluation contexts that lead to important decisions for the individuals, for instance the forensic context such as the assignment or the renewal of the firearms license, the fitness to drive, and the evaluation of parenting competences [73,74,75,76,77].

## Figures and Tables

**Figure 1 brainsci-10-00774-f001:**
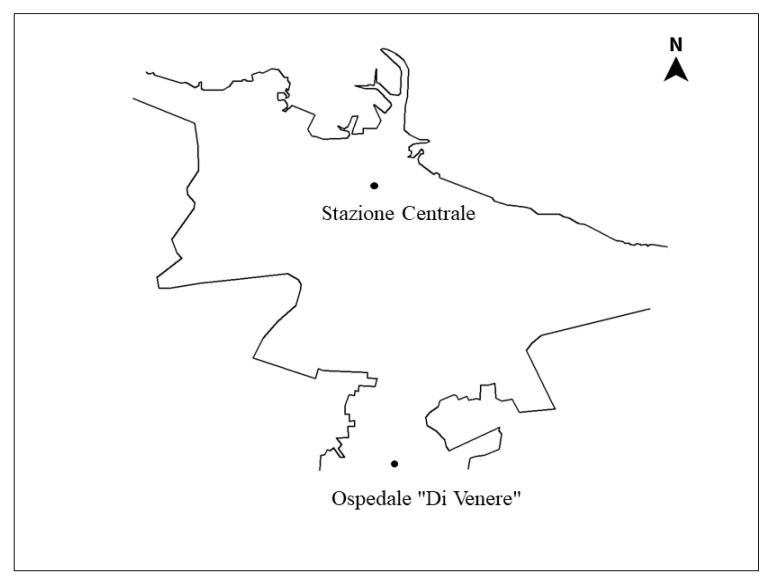
Semi-blind map with two fixed reference points and North.

**Figure 2 brainsci-10-00774-f002:**
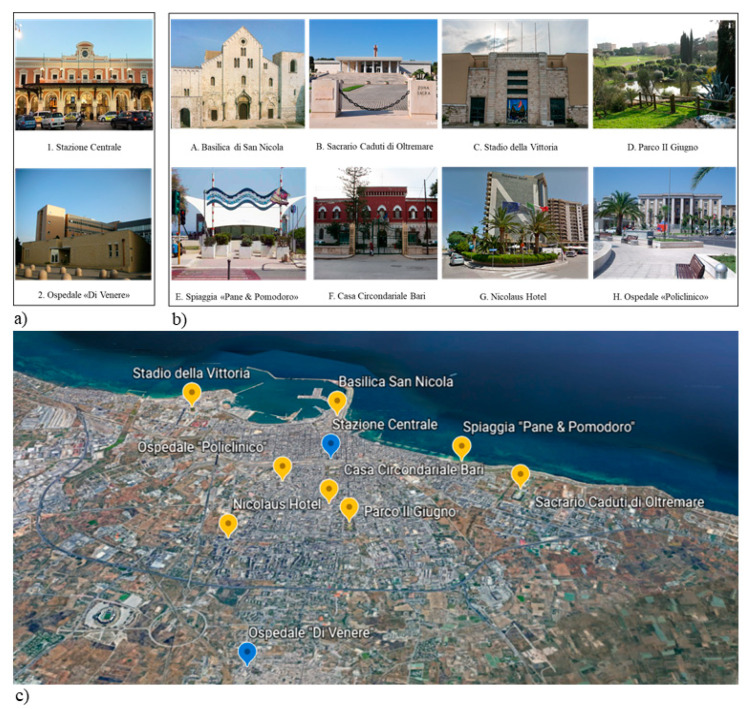
Map of Hometown, City of Bari: (**a**) pictures of fixed reference points; (**b**) pictures of landmarks to be placed on the semi-blind map; (**c**) the map with the expected positions and distances (Illustration free downloaded from Google Earth).

**Figure 3 brainsci-10-00774-f003:**
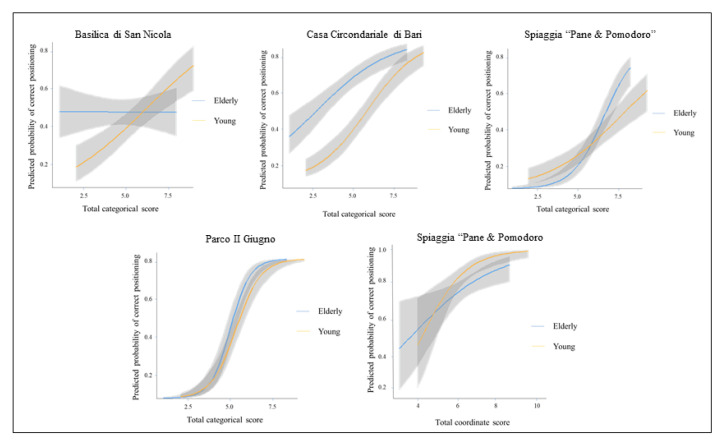
Item response curves of items reporting age DIF on categorical and coordinate scores.

**Figure 4 brainsci-10-00774-f004:**
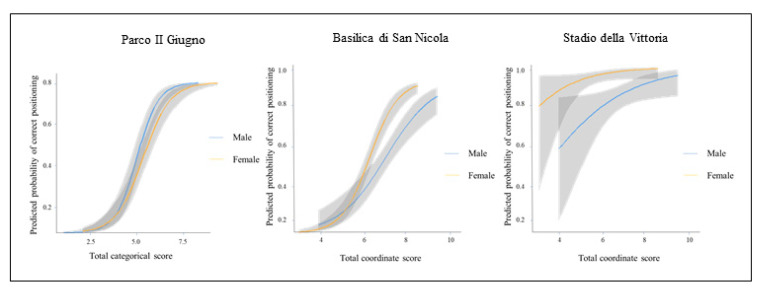
Item response curves reporting gender DIF on categorical and coordinate scores.

**Table 1 brainsci-10-00774-t001:** Means ± standard deviations for Categorical and coordinate scores are based on z transformations. Significant main effects of age-group and gender on demographical variables were obtained through *t* test. *** *p* < 0.001.

	YOUNG		ELDERLY			
	WOMEN	MEN	WOMEN	MEN	Group Age	Gender
	(*N* = 103)	(*N* = 100)	(*N* = 98)	(*N* = 99)		
Age, years	24 ± 6	23 ± 6	73 ± 7	73 ± 7	***	-
Education, years	16 ± 2	16 ± 1	11 ± 5	12 ± 5	***	n.s
Categorical score	0.23 ± 0.92	−0.31 ± 1.05	−0.15 ± 0.94	0.02 ± 1.02	n.s	n.s
Coordinate score	0.37 ± 0.88	0.08 ± 0.85	−0.56 ± 0.89	−0.03 ± 1.07	***	n.s

**Table 2 brainsci-10-00774-t002:** Number of DIF items in Categorical and Coordinate components, in Young and Elderly participants.

Effect Size (ZT)	YOUNG vs. ELDERLY	
	Category	Coordinate
A negligible	4	1
B moderate	0	0
C large	0	0

**Table 3 brainsci-10-00774-t003:** Binary logistic regression coefficients across age groups on categorical scores.

Item	Zumbo-Thomas	*p*-Value	χ^2^ Stat.	Δ*R*^2^	G*θ	Odd
Basilica di San Nicola	A	0.004	11.060	0.037	−0.353	0.700
Casa Circondariale Bari	A	<0.001	19.080	0.053	−0.197	0.820
Nicolaus Hotel	No flag	0.212	3.100	0.005	0.493	1.630
Spiaggia”Pane & Pomodoro”	A	0.003	11.480	0.041	0.601	1.825
Parco II Giugno	A	0.010	9.300	0.016	0.363	1.430
Ospedale “Policlinico”	No flag	0.242	2.840	0.005	0.006	1.005
Sacrario Caduti di Oltremare	No flag	0.089	4.850	0.010	0.322	1.370
Stadio della Vittoria	No flag	0.070	5.320	0.030	0.087	1.080

Note. Tests of significance conducted using: 2 degrees of freedom (χ^2^ significance threshold = 5.991); Δ*R*^2^: effect sizes scale; G*θ: interaction between the matching and grouping variables.

**Table 4 brainsci-10-00774-t004:** Binary logistic regression coefficients across age groups on coordinate scores.

Item	Zumbo-Thomas	*p*-Value	χ^2^ Stat.	Δ*R*^2^	G*θ	Odd
Basilica di San Nicola	No flag	0.823	0.390	0.001	−0.005	0.990
Casa Circondariale Bari	No flag	0.097	4.665	0.010	−0.228	0.790
Nicolaus Hotel	No flag	0.314	2.314	0.013	0.350	1.410
Spiaggia”Pane & Pomodoro”	A	0.001	13.530	0.067	−0.401	0.670
Parco II Giugno	No flag	0.257	2.720	0.007	−0.229	0.790
Ospedale “Policlinico”	No flag	0.287	2.495	0.006	−0.420	0.650
Sacrario Caduti di Oltremare	No flag	0.907	0.196	0.001	−0.218	0.800
Stadio della Vittoria	No flag	0.101	4.583	0.032	0.458	1.560

Note. Tests of significance conducted using: two degrees of freedom (χ^2^ significance threshold = 5.991); Δ*R*^2^: effect sizes scale; G*θ: interaction between the matching and grouping variables.

**Table 5 brainsci-10-00774-t005:** Number of DIF items in Categorical and Coordinate components, in women and men.

Effect Size (ZT)	WOMEN vs. MEN	
	Category	Coordinate
A negligible	1	2
B moderate	0	0
C large	0	0

**Table 6 brainsci-10-00774-t006:** Binary logistic regression coefficients across gender groups on categorical scores.

Item	Zumbo-Thomas	*p*-Value	χ^2^ Stat.	Δ*R*^2^	G*θ	Odd
Basilica di San Nicola	No flag	0.694	0.731	0.002	0.065	1.060
Casa Circondariale Bari	No flag	0.816	0.407	0.001	−0.066	0.940
Nicolaus Hotel	No flag	0.145	3.857	0.006	−0.333	0.710
Spiaggia”Pane & Pomodoro”	No flag	0.821	0.394	0.001	0.099	1.090
Parco II Giugno	A	0.032	6.861	0.013	0.581	1.780
Ospedale “Policlinico”	No flag	0.257	2.716	0.005	0.020	1.020
Sacrario Caduti di Oltremare	No flag	0.497	1.399	0.003	0.081	1.080
Stadio della Vittoria	No flag	0.511	1.342	0.007	0.198	1.200

Note. Tests of significance conducted using: two degrees of freedom (χ^2^ significance threshold = 5.991); Δ*R*^2^: effect sizes scale; G*θ: interaction between the matching and grouping variables.

**Table 7 brainsci-10-00774-t007:** Binary logistic regression coefficients across gender groups on coordinate scores.

Item	Zumbo-Thomas	*p*-Value	χ^2^ Stat.	Δ*R*^2^	G*θ	Odd
Basilica di San Nicola	A	0.049	6.041	0.015	−0.555	0.570
Casa Circondariale Bari	No flag	0.619	0.959	0.002	0.191	1.200
Nicolaus Hotel	No flag	0.935	0.135	0.001	0.087	1.080
Spiaggia “Pane & Pomodoro”	No flag	0.521	1.304	0.005	0.274	1.300
Parco II Giugno	No flag	0.231	2.927	0.007	−0.089	0.920
Ospedale “Policlinico”	No flag	0.246	2.808	0.006	−0.049	0.950
Sacrario Caduti di Oltremare	No flag	0.462	1.546	0.011	−0.586	0.560
Stadio della Vittoria	A	0.007	9.874	0.070	−0.150	0.860

Note. Tests of significance conducted using: 2 degrees of freedom (χ^2^ significance threshold = 5.991); Δ*R*^2^: effect sizes scale; G*θ: interaction between the matching and grouping variables.

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
