# Peer review of "Studying Individual Differences in Spatial Cognition Through Differential Item Functioning Analysis"

_brainsci, 2020, doi:10.3390/brainsci10110774_

Round 1
Reviewer 1 Report
This study addresses the application of the Differential Item Functioning Analysis (DIF) on a spatial task, namely the Landmark positioning on a Map (LPM) task, across age groups (young and elderly) and gender. In general results were satisfactory, in terms of categorical score no group differences emerged, whereas in terms of the overall coordinate score age differences occurred; younger were better than older participants.
I find the study well conducted, the use of the DIF in such a spatial task for the first time is remarkable. Authors also used a large sample size. Analyses sound good.
I have only some minor concerns:
1) give more details about the rational that is behind the study; authors stated that such an approach was used for the first time with spatial cognition, but I would suggest to add also some information about the practical/thoeretical rational.
2) I suggest to extend the part related to the implications of this study, in the discussion section (page 11, line 348). For example, which are the implications for the clinical practice? How can this approach deal with other important variables affecting spatial cognition/navigation, such as the cognitive style?
3) I suggest to add the following references:
a) Bocchi, A., Palermo, L., Boccia, M. Palmiero, M., D’Amico, S., & Piccardi, L. (2020). Object recognition and location: Which component of object location memory for landmarks is affected by gender? Evidence from four to ten year-old children. Applied Neuropsychology: Child, 9(1): 31-40. DOI: 10.1080/21622965.2018.1504218
b) Bocchi, A., Giancola, M., Piccardi, L., Palmiero, M., Nori, R., & D’Amico, S. (2018). How would you describe a familiar route or put in order the landmarks along it? It depends on your cognitive style! Experimental Brain Research, 236, 3121-3129. DOI: 10.1007/s00221-018-5367-3
Author Response
Reviewer #1:
This study addresses the application of the Differential Item Functioning Analysis (DIF) on a spatial task, namely the Landmark positioning on a Map (LPM) task, across age groups (young and elderly) and gender. In general results were satisfactory, in terms of categorical score no group differences emerged, whereas in terms of the overall coordinate score age differences occurred; younger were better than older participants. I find the study well conducted, the use of the DIF in such a spatial task for the first time is remarkable. Authors also used a large sample size. Analyses sound good. I have only some minor concerns:
Give more details about the rational that is behind the study; authors stated that such an approach was used for the first time with spatial cognition, but I would suggest to add also some information about the practical/thoeretical rational.
- Reply: Thank you very much for this comment. From line1 16 to line 126, we added the practical/thoeretical rational as follows:” In this view, the theoretical and practical relevance for using DIF analysis in the field of spatial cognition and in particular in spatial memory based on familiar topographical information is given by, for instance, the selection of items which can be representative and fair for many groups of people. Such selection could stimulate the construction of standardized tools, even within an ecological approach, to study spatial mental representations, where it could be difficult to control for previous learning processes, strength and accessibility of memory trace, and, as such for individual differences. DIF analysis allows to identify the differential functioning between groups. In this framework, it is then possible a) to remove the unfair landmarks with respect to different groups or b) to select landmarks which are appropriate for a specific group only. It is worthwhile to note that DIF analysis do not tell anything per se about the fairness or the biasedness of a test and, consequently, the eventual choice to exclude /substitute an item is demanded to the researcher.”
I suggest to extend the part related to the implications of this study, in the discussion section (page 11, line 348). For example, which are the implications for the clinical practice? How can this approach deal with other important variables affecting spatial cognition/navigation, such as the cognitive style?
- Reply: Thank you for this suggestion. It was done in the added conclusion
I suggest to add the following references:a) Bocchi, A., Palermo, L., Boccia, M. Palmiero, M., D’Amico, S., & Piccardi, L. (2020). Object recognition and location: Which component of object location memory for landmarks is affected by gender? Evidence from four to ten year-old children. Applied Neuropsychology: Child, 9(1): 31-40. DOI: 10.1080/21622965.2018.1504218; b) Bocchi, A., Giancola, M., Piccardi, L., Palmiero, M., Nori, R., & D’Amico, S. (2018). How would you describe a familiar route or put in order the landmarks along it? It depends on your cognitive style! Experimental Brain Research, 236, 3121-3129. DOI: 10.1007/s00221-018-5367-3 “b”.
- Reply: Thank you for this suggestion. It was done.
Reviewer 2 Report
This is an interesting paper. My concerns are as follows/
- The explanations in the Discussion and Conclusion section are addressed to uniform DIF. Meanwhile, the Basilica di san Nicolo item, for example, clearly exhibit non-uniform DIF when age groups are compared on categorical score (Figure 3). Why does the probability of the young to locate the item increase when the total categorical score increases, whereas the probability of the elderly not? Have you an explanation? Please, specify. There are other items in Figure 3 showing non-uniform DIF.
- The terms “latent ability variable” seems to be not a good one. Please consider the possibility of replacement by “latent ability measure” or “latent ability estimate” or something like.
- Lines 251, 259, 266, 272, 282: Figure 2 or Figure 3?
- Lines 301, 313, 321: Figure 3 or Figure 4?
Author Response
Reviewer #2:
This is an interesting paper. My concerns are as follows/
The explanations in the Discussion and Conclusion section are addressed to uniform DIF. Meanwhile, the Basilica di san Nicolo item, for example, clearly exhibit non-uniform DIF when age groups are compared on categorical score (Figure 3). Why does the probability of the young to locate the item increase when the total categorical score increases, whereas the probability of the elderly not? Have you an explanation? Please, specify. T
Reply: Thank you for this suggestion. The answer to your questions is reported from line 364 to line 370 of the manuscript. The explanation we gave was related to landmark placement process, that seemed to be more difficult for the elderly compared to the young.
There are other items in Figure 3 showing non-uniform DIF.
- Reply: Thank you for this suggestion. We explain the Uniform type of DIF for Item#2: Casa Circondariale di Bari, as emerged from figure 3, as follows: “The age DIF test is statistically significant χ 2 (2) = 19.08, p < 0.001. The ΔR2 = 0.053 is a small effect by ZT criterion. Both the significance test and the effect size suggested age DIF. As emerged from Table 3 the young were uniformly 1.2 times more likely to correctly place item#2 than the elderly.”
The terms “latent ability variable” seems to be not a good one. Please consider the possibility of replacement by “latent ability measure” or “latent ability estimate” or something like.
- Reply: Thank you for this suggestion. We used latent ability measure.
Lines 251, 259, 266, 272, 282: Figure 2 or Figure 3? Lines 301, 313, 321: Figure 3 or Figure 4?.
- Reply: Thank you for this suggestion. We checked the numbers.